# Co-Supplementation of Policosanol and Banaba Leaf Extract Exhibited a Cooperative Effect Against Hyperglycemia and Dyslipidemia in Zebrafish: Highlighting Vital Organ Protection Against High-Cholesterol and High-Galactose Diet

**DOI:** 10.3390/ijms26167669

**Published:** 2025-08-08

**Authors:** Kyung-Hyun Cho, Sang Hyuk Lee, Yunki Lee, Ashutosh Bahuguna, Ji-Eun Kim, Cheolmin Jeon

**Affiliations:** Raydel HDL Research Institute, Medical Innovation Complex, Daegu 41061, Republic of Korea

**Keywords:** oxidative stress, paraoxonase, apoptosis, body weight, fatty liver, ovaries, senescence

## Abstract

The efficacy of *Lagerstroemia speciosa* (banaba) leaf extract (BLE), policosanol (POL), and their combination (BLE+POL) was evaluated in zebrafish (*Danio rerio*) against high cholesterol (HC)- and galactose (HG)-induced metabolic stress and organ toxicity. After 12 weeks of dietary intervention, BLE+POL significantly reduced HC+HG-augmented weight gain and improved hepatic and nephromegaly. Compared with BLE or POL alone, the combined intake of BLE+POL more effectively alleviated dyslipidemia and blood glucose levels. Likewise, BLE+POL effectively reduced blood malondialdehyde (MDA), aspartate aminotransferase (AST), and alanine aminotransferase (ALT) levels and boosted plasma sulfhydryl content, ferric ion reduction ability (FRA), and paraoxonase (PON) activity. Histological outcomes suggest that BLE+POL has higher efficacy than either BLE or POL in mitigating HC+HG-induced fatty liver changes, hepatic inflammation, kidney senescence, and reactive oxygen species (ROS) production. Consistently, BLE+POL augmented the spermatozoa counts in the testes, enhanced mature vitellogenic oocytes in ovaries, and protected them from the HC+HG-induced oxidative stress. Compared with either BLE or POL, a combined intake of BLE+POL displayed a superior effect in inhibiting the apoptosis and accumulation of lipid peroxidation species 4-hyrdoxynonenal (4-HNE) in the brain. A combined intake of BLE+POL exhibited a pronounced impact than the BLE and POL alone and can be utilized as an effective formulation to counteract the HC+HG-induced events.

## 1. Introduction

Dyslipidemia is a metabolic disorder commonly marked by elevated blood levels of total cholesterol (TC), triglycerides (TGs), and low-density lipoprotein cholesterol (LDL-C), along with decreased concentrations of high-density lipoprotein cholesterol (HDL-C) [1]. A lethargic lifestyle, poor food habits, and smoking are the major contributors to the development of dyslipidemia [1,2]. Substantial evidence indicates that dyslipidemia negatively impacts the function of several vital organs (like the liver, kidneys, ovaries, and testes) [3,4,5,6]. Furthermore, the notable influence of dyslipidemia has been well established with coronary heart disease [1,7]. Similarly, high galactose consumption favors various adverse events mainly provoked by the formation of advanced end glycation products (AGEs), leading to premature aging and impaired functionality of various organs, including the liver, kidney, eyes, and brain [8,9,10,11,12]. Collectively, a high-cholesterol and high-sugar dietary regime provides a robust experimental model for inducing dyslipidemia, senescence, hyperglycemia, and impaired functionality of various organs closely mimicking human pathophysiological conditions [13].

Different synthetic compounds and drugs have been explored aiming at dyslipidemia; however, their long-term consumption has often emerged with certain adverse effects [14,15]. In contrast, many natural compounds and extracts have shown an impact on dyslipidemia [16] and other metabolic diseases [17] and are generally considered safe. Among the variety of natural compounds, policosanol has gained considerable acceptability in countering dyslipidemia [18,19]. Studies have even revealed that policosanol, in combination with classical lipid-lowering drugs (statins), effectively minimized the statins-induced side effects [20,21]. Policosanol is a generic term used for the mixture of long-chain aliphatic alcohols (LCAA, C_24_-OH ~ C_34_-OH) that can be extracted from a variety of source materials of animal and plant origins [19]. Based on the source material, climatic conditions, and extraction method, the composition of policosanol varied substantially, ultimately impacting its functionality. The accumulating literature showed the functional disparity of policosanol towards dyslipidemia; nevertheless, most studies confirmed policosanol’s functionality to counter dyslipidemia [19]. Nonetheless, over 25 countries have officially approved policosanol as an effective and safe cholesterol-lowering agent [22]. Besides the effect on dyslipidemia, policosanol intake improved blood pressure [18,23] lipid peroxidation markers [23], and blood glucose levels [19,24].

*Lagerstroemia speciosa* (commonly referred as banaba) is a medicinal plant extensively grown in South Asian countries and is recognized for its wide range of biological activities including antioxidant, anti-inflammatory, and analgesic properties [25,26]. Notably, the banaba leaves (due to their major functional component, corosolic acid) gained huge attention as an antidiabetic agent through its influence on glucose metabolism, cellular glucose uptake, and improvement of insulin secretion and sensitization [25,27]. Moreover, banaba/corosolic acid showed an inhibitory effect on the α-amylase and α-glucosidase, thus modulating postprandial glucose levels efficiently [28]. 

Despite several individual studies embarking on the functionality of banaba [25,26,28] and policosanol [19,23], only a few preliminary studies were conducted on their combined formulation. One such study commenced by our research group decoded the comparative effect of banaba leaf extract, policosanol, and their combination against streptozotocin (STZ)-induced diabetic tissue regeneration and oxidative stress in adult zebrafish [29]. Herein, in the extension of previous studies, we fed zebrafish (*Danio rerio*) with banaba leaf extract (BLE, final 0.1%, *w/w*), policosanol (POL, final 0.1%, *w/w*), or the combination of BLE+POL (final 0.1%, *w/w* each) for 12 weeks under the chronic influence of high cholesterol (HC, final 4% *w/w*)+galactose (HG, final 30% *w/w*) diet and investigated their influence on mortality and body weight change. In addition, the comparative effect of BLE, POL, and BLE+POL was examined on dyslipidemia, blood glucose level, oxidative stress markers, antioxidant capacity, and histological alterations in the liver, kidney, brain, and reproductive organs of zebrafish.

In their natural habitat, zebrafish intake small plankton and algae as food. However, under controlled laboratory conditions, their diet can be modified by adding specific chemicals and supplements to induce a variety of events, including dyslipidemia [29,30,31,32,33]. In this study, zebrafish were deliberately administered an HC+HG diet to induce dyslipidemia and glycemia, simulating physiological events similar to those observed in lifestyle and disease-related conditions such as type II diabetes. This study aims to serve as a fundamental platform for evaluating therapeutic efficacy (of BLE, POL, and BLE+POL) relative to multiple pathophysiological contexts, including dyslipidemia induced by food habits.

Notably, zebrafish serve as a valuable model for biomedical research owing to their strong genetic resemblance to humans [30] and their ability to mimic human disease mechanisms. Specifically, their lipid metabolism pathway, which includes homologous receptors, enzymes, and lipoproteins, closely aligns with that of humans [31]. Therefore, compounds or extracts that show efficacy in zebrafish may offer promising translational potential for human application.

## 2. Results

### 2.1. Survivability and Changes in Body Weight

Survivability remained at 100% across all groups during 12-week interventions of different diets. Contrary to survivability, body weight was substantially altered following the different diets over time (0–12 weeks) (Figure 1A,B).

The repeated analysis of variance (ANOVA) test examined the statistical difference among the effects of varied time points, diets, and combinations of diets with time. The repeated ANOVA test following different multivariate analysis (e.g., Wilks’s lambda, Roy’s largest root, Pillai’s trace, and Hotelling’s trace) showed a significant (F = 159.6, *p* = 0.00) effect of 0-, 8-, and 12-weeks’ time on the body weight enhancement across all the groups (Figure 1A). The data on the body weight between the different diets showed the maximum attainment of 67.2% body weight in the HC+HG group, followed by BLE (60.2%), POL (48.4%), BLE+POL (34.9%), and the least in the ND (27.7%)-supplemented group compared with the body weight of week 0. The multivariate analysis of the body weight changes in the 12 weeks showed a significant 48.1% (*p* < 0.05) reduction in body weight in the BLE+POL-supplemented group compared with the body weight of the HC+HG-supplemented group. In contrast, no significant difference in body weight changes was observed for BLE and POL compared with the HC+HG group (Figure 1B). The interaction of time with different diets assessed by repeated multivariate analysis revealed a significant effect (F = 7.0, *p* = 0.00) of these interactions on the body weight changes of zebrafish. Notably, throughout the experimental period (week 0, 8, and 12), food consumption remained consistent across all groups (approximately 95–100%), suggesting that different dietary formulations did not influence the zebrafish feeding preference or appetite.

Morphometric analysis indicated a significant (*p* < 0.001) decline in the body length (BL) to body depth (BD) ratio (BL/BD) in the HC+HG-consumed group compared with the ND (control) group (Figure 1C,D). The combined supplementation of BLE+POL effectively mitigated the HC+HG-induced morphometric parameters, reflected by a considerably lower (*p* < 0.05) BL/BD ratio compared with the HC+HG group. In contrast, a non-significant effect of individual supplementation of BLE and POL was noticed on the HC+HG alleviated BL/BD ratio.

### 2.2. Changes in Morphology and Organ Weights

Morphological assessment of the liver indicates hepatomegaly in the HC+HG-consumed group, with a significant increase in liver weight (average ~17.8 mg) that is significantly 3.4-fold (*p* < 0.001) higher than the liver weight quantified in the ND group (control) (Figure 2A). Co-supplementation of either BLE or POL showed a substantial effect on HC+HG-induced hepatomegaly and liver body weight, which was indicated by a significantly reduced liver weight. While compared with the individual effect, the combined supplementation of BLE+POL displayed a considerably better effect, marked by 3.3-fold, 2.3-fold, and 2.2-fold reduced liver weight compared with HC+HG-, BLE-, and POL-consumed groups. Likewise, BLE+POL also significantly protects the kidneys from HG+HG nephromegaly (Figure 2B). Individually, both BLE and POL substantially impact HC+HG-induced kidney morphology and weight; however, these effects are inferior compared with the joint impact of BLE+POL.

Intriguingly, no significant effect was noticed between the groups (ND, HC+HG, BLE, POL, and BLE+POL) on the brain and ovary morphology and weight (Figure 2C,D). In contrast, testes morphology and weight were substantially altered by the consumption of HC+HG, and the supplementation of BLE, POL, and BLE+POL significantly improved these changes (Figure 2E). However, when comparing BLE, POL, and BLE+POL, a non-significant change in the testes weight was observed between these treatments.

### 2.3. Blood Lipoprotein Profile and Glucose Levels

Compared with the TC (149.1 ± 10.4 mg/dL), TG (87.8 ± 9.8 mg/dL), and HDL-C (111.5 ± 5.3 mg/dL) levels of the ND group (control), elevated TC (220.9 ± 13.2 mg/dL) and TG (137.6 ± 4.1 mg/dL) levels and diminished HDL-C (65.1 ± 6.4 mg/dL) levels were noticed in response to the HC+HG supplementation. The HC+HG-disturbed blood lipoprotein parameters were significantly prevented by the supplementation of BLE and POL (Figure 3A–C). As compared with HC+HG, considerably reduced TC (197.1–190.8 mg/dL) and TG (113.2–110.3 mg/dL) levels and elevated HDL-C (84.3–88.9 mg/dL) levels were quantified in the BLE- and POL-consumed group. The combined supplementation of BLE+POL was found to be more effective to reduce TC (177.2 ± 7.6 mg/dL) and TG (96.2 ± 7.4 mg/dL) and elevate the HDL-C (97.3 ± 9.4 mg/dL) levels, which were noticed to be ~11–7%, 15–12%, and 15–9.4% better than the individual effect exerted by BLE and POL, respectively. Consistently, the lowest percentage of HDL-C/TG (29.6%) and a heightened ratio of TG/HDL-C (2.2) were noticed in the HC+HG-supplemented group, which was significantly reverted by the consumption of BLE and POL (Figure 3D,E). Compared with the individual supplementation BLE and POL, a combination of BLE+POL showed a ~28–17% elevated HDL-C/TG (%) and a ~26–20% reduction in the TG/HDL-C ratio.

The HC+HG-elevated blood glucose level (70.4 ± 1.1 mg/dL) was significantly reduced by the co-supplementation of BLE (59.2 ± 1.2 mg/dL) and POL (62.1 ± 1.5 mg/dL) (Figure 3F). In combination (BLE+POL), it was found to be ~11% and ~15% more effective to reducing the blood glucose level than the individual supplementation of BLE and POL, respectively. Notably, when compared with the blood glucose level of the ND group (49.2 ± 1.4 mg/dL), a non-significant difference in the glucose level was observed in the BLE+POL-supplemented group, suggesting the effectiveness of BLE+POL to diminish the HC+HG-elevated glucose level to the normal basal level.

### 2.4. Oxidative Stress, Antioxidant, and Hepatic Function Variables in Blood

The HC+HG-supplemented groups showed the highest amount of plasma MDA (9.5 ± 0.5 μM), which was significantly 1.9-fold higher than the MDA level qualified in the ND (control) group. Supplementation of BLE, POL, and BLE+POL effectively reduced HC+HG-elevated plasma MDA level (Figure 4A). Contrary to the MDA, the least sulfhydryl content was quantified in the HC+HG group (8.4 ± 0.3 nmol/mg protein), which was increased by ~26% and 30% by the supplementation of BLE and POL, respectively (Figure 4B). Compared with the individual intake of BLE and POL, the combined supplementation (BLE+POL) showed a 22% and 16.5% higher plasma sulfhydryl content, respectively.

The lowest plasma FRA (328.5 ± 14.5 μM) and PON (4.9 ± 0.6 μU/L/min) activity was noticed in the HC+HG-consumed group, which was significantly augmented by 18–25% and 34–46% by the supplementation of BLE and POL (Figure 4C,D). The BLE+POL combined supplementation emerged with substantially higher PON (8.4 ± 0.7 μU/L/min) and FRA (442.7 ± 8.6 μM) ability, which was ~25–16% and ~14–8% higher than the activities observed in the BLE and POL individually supplemented groups, respectively.

The HC+HG supplementation substantially elevated the plasma AST (531.8 ± 34.1 IU/L) and ALT (421.2 ± 37.1 IU/L) levels, which were significantly (*p* < 0.001) 1.8-fold (300.8 ± 6.1 IU/L) and 2.3-fold (179.7 ± 9.9 IU/L) higher than their respective levels noticed in the ND control group. The HC+HG-elevated AST and ALT levels were ~18–22% and ~19–24% reduced following the consumption of BLE and POL (Figure 4E,F). The combined supplementation of BLE+POL showed a much-pronounced effect, reflected by ~34% and 42% reduced AST and ALT levels compared with the HC+HG-supplemented group, which is significantly ~20–18% and ~28–24% lower than the AST and ALT levels observed in the BLE- and POL-consumed group.

### 2.5. Liver Histology

The highest neutrophil count was noticed in the HC+HG groups, which was significantly reduced by 39% and 73% following the BLE and POL supplementation (Figure 5A,B,F). In contrast to the individual supplementation, a combined supplementation (BLE+POL) showed a much-pronounced effect, highlighted by 80%, 69%, and 32.5% lower neutrophil counts than the HC+HG-, BLE-, and POL-supplemented groups, respectively.

ORO staining revealed the highest lipid accumulation (18.5% ORO-stained area) in the HC+HG-consumed group, which was significantly 2-fold, 6-fold, and 25-fold alleviated by the consumption of BLE, POL, and the combination of BLE+POL, respectively (Figure 5C,G). Compared with the individual supplementation of BLE and POL, the combined supplementation of BLE+POL showed a significant 12.8-fold and 3.9-fold reduced ORO-stained area, respectively.

Similar to the ORO staining, the highest IHC-stained area (20.7%), corresponding to IL-6 production, was quantified in the HC+HG-consumed group, which was significantly reduced to 36%, 57%, and 72% by the supplementation of BLE, POL, and BLE+POL, respectively (Figure 5D,E,H). A significant 2.3-fold and 1.6-fold diminished IL-6-stained area compared with the BLE- and POL-consumed group was noticed in the BLE+POL-supplemented group, testifying to the higher efficacy of BLE+POL to inhibit IL-6 compared with their individual supplementation.

### 2.6. Dihydroethidium (DHE) Fluorescence Staining and Senescent-Associated β-Galactosidase Staining 

The DHE-staining and SA-β-gal staining revealed the highest ROS production and cellular senescence in the hepatic tissue from the HC+HG-consumed group (Figure 6A–D). The HC+HG-induced ROS production and cellular senescence was effectively prevented by the supplementation of BLE, POL, and BLE+POL, which reflected 1.4-fold, 2.5-fold, and 3.1-fold reduced DHE-stained area and 1.8-fold, 3.6-fold, and 3.8-fold reduced SA-β-gal-stained area, respectively, compared with the HC+HG groups. In contrast to the individual supplementation of BLE, the combined supplementation of BLE+POL showed a significant 2.2-fold and 2.1-fold higher efficacy to inhibit the ROS production and cellular senescence, respectively.

### 2.7. Kidney Histology

The kidney section of the HC+HG group appeared with disturbed and unorganized distal tubes (DTs) and proximal tubes (PTs) and the prevalence of dilated tubular lumen (indicated by the red arrow) with the presence of luminal debris (indicated by the blue arrow) (Figure 7A). Moreover, the occasional presence of a dark blue-stained (basophilic cluster) area related to the new nephron generation was noticed (indicated by green arrow). The supplementation of BLE and POL substantially protects the kidney from the HC+HG-induced pathological changes; however, the occasional presence of dilated tubular lumen and luminal debris was noticed. In contrast, BLE+POL effectively mitigates the HC+HG-induced kidney damage, reflected by a properly arranged tubular structure that was mostly free from the dilated lumen and luminal debris.

The highest DHE fluorescence intensity corresponding to kidney ROS generation was noticed in the HC+HG-supplemented group, which was reduced to 29.6% and 65.8% following the consumption of BLE and POL (Figure 7B,D). Compared with the individual effect of BLE and POL, the combined supplementation of BLE+POL displayed a significant 3.8-fold and 1.8-fold higher inhibitory effect on DHE fluorescence intensity.

The highest SA-β-gal-positive cells (19.3%) corresponding to cellular senescence were observed in the HC+HG-consumed group, which was significantly prevented by the consumption of BLE, POL, and BLE+POL (Figure 7C,E). However, the combined supplementation of BLE+POL showed a ~34% and 15% better effect on inhibiting cellular senescence as compared with the consumption of BLE and POL alone.

### 2.8. Histology of Testicular Tissue

The H&E staining of the testes of the HC+HG-supplemented group revealed a reduced number of spermatozoa (6.1%) with an elevated interstitial space between the seminiferous tubules (Figure 8A,B,D,E). The HC+HG-elevated interstitial space between the seminiferous tubules is significantly minimized to 12.6%, 17.2%, and 25.8% following BLE, POL, and BLE+POL supplementation. Similarly, BLE, POL, and BLE+POL showed 9.4%, 13.9%, and 15.1% spermatozoa area, corresponding to 1.6-fold, 2.3-fold, and 2.5-fold higher areas than the spermatozoa area quantified in the HC+HG group, respectively.

The DHE and SA-β-gal staining revealed a massive ROS generation and cellular senescence in testicular tissue in response to HC+HG supplementation (Figure 8B,C,F,G). Individually, both BLE and POL consumption failed to significantly prevent the HC+HG-induced ROS generation. However, both BLE and POL effectively reduced the senescence, evident by a 1.7-fold and 3-fold reduction in SA-β-gal-positive cells compared with the HC+HG group. Contrary to the individual supplementation of BLE and POL, a combined supplementation (BLE+POL) effectively protected the HC+HG-triggered ROS generation and cellular senescence, highlighted by a significantly 2.8-fold and 3.8-fold reduction in DHE fluorescence intensity and SA-β-gal-positive cells, respectively. When compared with the effect of BLE and POL, a 2.2–1.9-fold reduction in DHE fluorescence intensity and 2.3–1.3-fold SA-β-gal-positive cells were quantified in the BLE+POL-supplemented group, attesting the synergistic implication between BLE and POL when consumed in combination.

### 2.9. Histology of Ovarian Tissue

The H&E staining of the ovary revealed the high prevalence of pre-vitellogenic oocytes (96.8%) and decreased counts of early (1.2%) and mature (1.9%) oocyte counts in the HC+HG-supplemented group (Figure 9A,E–G). BLE, POL, and BLE+POL supplementation effectively reduced the HC+HG-elevated pre-vitellogenic oocytes and augmented early and mature oocyte counts. Strikingly, BLE+POL showed a significant 2.7-fold and 1.6-fold higher amount of mature vitellogenic oocytes compared with BLE or POL alone, reflecting the synergistic implication of combined supplementation to protect the ovary.

A high amount of lipid accumulation (23.7% ORO-stained area) was observed in the HC+HG-consumed group, which was significantly reduced to 16.2% and 11.4% ORO-stained area by the supplementation of POL and BLE+POL (Figure 9B,H). A non-protective effect of BLE was noticed in the lipid accumulation in the ovaries. The DHE and SA-β-gal staining revealed a 1.5–2.4 fold reduced and 1.8–3.7-fold reduction in DHE fluorescence intensity and SA-β-gal-positive cells in the BLE- and POL-supplemented groups, compared with the HC+HG group, depicting the impact of BLE and POL against HC+HG induced ROS generation and cellular senescence (Figure 9C,D,I,J). A combined supplementation of BLE+POL showed the least DHE fluorescence intensity, which was significantly 2.2-fold and 1.4-fold lower than the DHE fluorescence intensity observed in BLE- and POL-supplemented groups. Likewise, least SA-β-gal-positive cells (1.3%) were detected in the BLE+POL supplemented group, which accounted for 2.6-fold and 1.2-fold lower counts than the SA-β-gal-positive cells detected in the BLE- and POL-supplemented group.

### 2.10. Histological Analysis of the Brain

The H&E staining of the brain revealed no marked histological differences concerning the hyperemia, vacuolation, and mononuclear cells, with clear zone in tectum opticum (TeO) and periventricular gray zone (PGZ) among the HC+HG, BLE, POL, and BLE+POL groups (Figure 10A,B and Appendix A). On the contrary, DHE and AO fluorescence staining revealed a massive ROS production and apoptosis in the HC+HG group, mainly around the TeO and PGZ region (Figure 10C,D,E,J,K). Supplementation of BLE and POL effectively inhibits the ROS production and apoptosis, as evidenced by 1.6–2-fold reduced DHE and 1.7–1.9-fold reduced AO fluorescence intensities compared with the respective intensities observed in the HC+HG group. Compared with the effect of either BLE or POL, the combined supplementation of BLE+POL showed a significant ~1.8–1.4 fold and ~2–1.8-fold higher reduction in DHE and AO fluorescence intensities.

The IHC staining revealed the higher accumulation of 4-HNE around the vascular lacuna of area postrema (Vas) below the tectal ventricle (Tev) in the HC+HG-supplemented group, which was significantly 2.1-fold, 2.9-fold, and 5.2-fold reduced by the supplementation of BLE, POL, and BLE+POL, respectively (Figure 10F,L). However, compared with either BLE or POL, BLE+POL displayed a significant 2.6-fold and 1.8-fold reduced 4-HNE accumulation, testifying to the higher efficacy of the BLE+POL combination to prevent the accumulation of 4-HNE in the brain.

The SA-β-gal staining revealed the high accumulation of blue-colored senescent-positive cells (2.9%) in the HC+HG section (Figure 10G–I,M). Significantly reduced SA-β-gal-positive cells (~1.5–1.2%) that accounted for ~1.9–2.4-fold lower senescence than the HC+HG group were observed in the BLE-, POL-, and BLE+POL-supplemented groups. Interestingly, a non-significant difference in the inhibition of cellular senescence was noticed between BLE-, POL-, and BLE+POL-supplemented groups.

## 3. Discussion

High-cholesterol diet is responsible for the induction of numerous adversities and the onset of many health implications [34]. Likewise, high sugar intake also severely causes oxidative stress, inflammation, premature aging, and damage to a variety of organs [35,36,37]. In combination, high cholesterol and galactose lead to obesity, metabolic stress, and resemble the pathophysiology and progression of the human diseases [13]. In the present study, 12 weeks’ intake of HC+HG leads to body weight enhancement, which was significantly protected by the intake of BLE+POL. A non-significant effect for either BLE or POL was detected on the management of the HC+HG-induced body weight. The effect of POL on body weight management has been described [23,38]; similarly, the BLE has also been reported for its anti-obesity effect [25]. However, under the chronic conditions of HC+HG, individually, BLE and POL failed to establish any significant effect on body weight. However, when used in combination (BLE+POL), a reduced body weight was observed, suggesting a BLE and POL cooperative implication towards HC+HG-induced body weight management.

The effect of BLE [39] and POL [23] has been documented as an anti-dyslipidemic effect. Particularly, the role of POL has been extensively reported to regulate the 3-hydroxy-3-methyl-glutaryl-coenzyme A reductase (HMG-CoA) (a rate limiting enzyme for the cholesterol biosynthesis) [40], inhibition of cholesteryl ester transfer protein (CETP) activity, and upregulation of the expression of apolipoprotein A-I (apoA-I), thus substantially modulating the plasma lipoprotein profile [41]. Herein, the HC+HG-elevated TC and TG levels and diminished HDL-C levels were substantially reverted by the consumption of BLE and POL which is consistent with the earlier reports highlighting the positive effect of BLE [39] and POL [23] on the plasma lipid profile. Nevertheless, the study outcome revealed that combined (BLE+POL) use has a significantly better impact than either BLE or POL to mitigate the HC+HG-induced dyslipidemia, testifying the BLE+POL interaction in a cooperative manner to manage the dyslipidemia effectively.

An evident adverse effect of oxidative stress, high-fat diet, and galactose on insulin secretion and resistance has been described, eventually impacting the blood glucose level [42,43]. Herein, a 12-week intake of a HC+HG diet augmented the blood glucose level, which was effectively reduced by the consumption of either BLE or POL. The role of BLE (due to corosolic acid) has been well described to modulate adenosine monophosphate-activated protein kinase (AMPK) and the Akt pathway, resulting in the modulation of insulin secretion and sensitization and a consequent effect on the blood glucose level [28,44]. Moreover, BLE also has an inhibitory effect against α-amylase and α-glucosidase, thus regulating the post-prandial glucose level [28]. Also, studies document the affirmative effect of policosanol (a mixture of LCAA) on diabetes [19] and the blood glucose level of zebrafish and human subjects [24]. The present study emphasizes that BLE+POL combination had a significantly better hypo-glycemic effect than either of BLE and POL, attesting the cooperative interaction between BLE and POL to manage the HC+HG-induced blood glucose level.

High-sugar intake has been shown to induce ROS production, oxidative stress, inflammation, and redox imbalance through multiple mechanisms, including the formation of advanced glycation end products (AGEs) [45]. Herein, the higher MDA level and diminished plasma sulfhydryl content was noticed in response to 12 weeks of HC+HG consumption, depicting the elevated oxidative variables in the blood. Additionally, a compromised antioxidant reflected by diminished plasma FRA and PON activity was observed in the HC+HG-supplemented group. The HC+HG-disturbed oxidative variables and antioxidant parameters are substantially reverted by the consumption of BLE and POL, though when they were used in combination (BLE+POL), a significantly better effect was observed than their individual supplementation. A strong antioxidant effect of BLE (due to corosolic acid) has been described that directly scavenges free radicals [46] and modulates Nrf2 expression [47]; an important regulator of cellular antioxidants like superoxide dismutase (SOD) and catalase [48] is behind the management of HC+HG-elevated plasma oxidative variables and antioxidant parameters. Similarly, POL antiglycation and an ROS inhibitory effect and positive effect on PON activity [23,49] has been documented that prevents oxidative damage against external stress. Herein, we believe that the distinct activities of BLE (direct radical scavenging and cellular antioxidant activity) and POL (antiglycation and PON activity) worked in a cooperative manner, which leads to much better protection against the HG+HG-disturbed plasma oxidative variable and antioxidant parameters.

The HC diet has been shown to damage the liver and provoke fatty liver changes. Also, high intake of galactose leads to oxidative stress, secretion of inflammatory cytokines like TNF-α, IL-6, and IL-10, and the induction of hepatic senescence [8]. Following 12 weeks of supplementation of HC+HG, a notably high number of neutrophil counts, fatty liver changes, massive production of IL-6, ROS generation, and senescent-positive cells in the liver were observed. The HC+HG-induced hepatic damage is substantially prevented by the BLE and POL supplementation, which was significantly improved when used in combination (BLE+POL). BLE’s role as an antioxidant and as an anti-inflammatory agent guided the positive incidents of liver protection against the HC+HG-induced events. It has been well established that BLE (due to corosolic acid) can directly scavenge the free radicals [46], modulate the cellular expression of antioxidants [47], and inhibit NFκB nuclear translocation [28], thus effectively preventing the oxidative stress and inflammation [28]. Similarity, POL’s effect to inhibit ROS generation, IL-6 production, and cellular senescence has been documented [49]. In addition, both POL (due to hexacosanol) and BLE (due to corosolic acid) are modulators of autophagy [40,50], which is considered a key cellular event towards the inhibition of fatty liver [40]. It is apparent that in combination, BLE+POL, due to their distinct functionality, interacted in a cooperative manner leads to a substantial higher liver protection than the individual supplementation to ameliorate the HC+HG-triggered hepatic damage. The least senescence in the BLE+POL group is attributed by the reduced ROS level in this group, as the ROS-induced oxidative stress has been recognized as the key instigator of senescence [51].

Substantial kidney and reproductive organ damage was noticed in the HC+HG-supplemented group, i.e., in accordance with the earlier reports describing the kidney and reproductive organ damaging effect of dyslipidemia [52,53,54] and intake of high galactose [55,56,57]. HC causes kidney damage by the induction of oxidative stress; likewise, galactose favors the AGE formation that generated oxidative stress [58] and leads to kidney damage. In response to the HC+HG diet, a higher ROS generation was observed in the kidney, ovaries and testes, which were effectively inhibited by the supplementation of BLE and POL; however, the most impactful effect was observed in association with BLE+POL. As oxidative stress and generation of AGE are the key contributors for kidney and reproductive organ damage; therefore, a substance having antioxidant [59] and antiglycation [55] activity have a substantial effect on these organs’ protection. Similarly, antioxidants have proved effective to protect the testes [60] and ovary [61] damage from stress. As in the combination (BLE+POL), the antioxidant activity of BLE [26,46,47] and the antiglycation nature of POL [49,62] worked in a close cooperative manner, resulting in better kidney and reproductive organ protection against the HC+HG-induced changes.

A detrimental effect of galactose on the brain via accumulation of AGE [63,64], generation of ROS [65], apoptosis, senescence, and disturbed antioxidant activity [66] has been documented. Similarly, we have noticed a high prevalence of ROS, particularly the high accumulation of lipid peroxidation product 4-HNE in the brain of the HC+HG-consumed group. The accumulation of 4-HNE has been recognized to cause severe oxidative stress and instigate neuronal cell death via caspase-3 mediate apoptosis [67]. The antioxidants have a substantial inhibitory effect on the accumulation of 4-HNE and subsequent adverse events. The better brain protective effect was noticed in BLE+POL group compared with the individual supplementation. The substantial anti-apoptotic, ROS inhibition effect of POL [49], and BLE’s antioxidant, anti-apoptotic [28,46,47] properties worked in a cooperative manner, leading to brain protection against HC+HG-induced events. In addition to the direct effect of BLE+POL on the brain, a substantial protective effect of BLE+POL on the liver and kidney health is also responsible for better brain health. This notion is in accordance with studies suggesting a deep association between liver–brain health [68] and kidney–brain health [69].

Despite the several beneficial effects of BLE+POL on the HC+HG-induced metabolic stress and associated organ damage, further investigation is warranted to assess its impact on blood cellular components and vascular plaque formation to establish the broad applicability of BLE+POL to mitigate the HC+HG-induced detrimental effects.

## 4. Materials and Methods

### 4.1. Materials

Ethanolic extract of *Lagerstroemia speciosa* (banaba) leaves was procured from Umalaxmi Organic Pvt. Ltd. (Jodhpur, Rajasthan, India). Cuban policosanol, extracted from sugarcane wax, which contains a unique mixture of eight long-chain aliphatic alcohols (LCAA, C_24_–C_34_), was provided by Raydel^®^ Pty, Ltd. (Thornleigh, NSW, Australia). A detailed product sheet and the specifications of banaba leaf extract (batch no. UO/LSD-1825/02/23-24) and policosanol (batch no. 310030324) are provided in Appendix A. Unless otherwise stated, all the others chemical and reagents were of analytical grade and used as supplied. A list of the chemicals and reagents is provided in Appendix A.

### 4.2. Zebrafish Culturing

Adult wild-type AB stains of zebrafish (~12 weeks of age) were cultured under the alternative photoperiod of dark (10 h) and light (14 h) in a water tank with a circulating water supply. The temperature of the water and the culture room was maintained at 28 °C. The standard guidelines of the Animal Care Committee and Use adopted by the Raydel Research Institute (code of approval RRI-23-007, date of approval 27 July 2023) were strictly followed during the zebrafish husbandry. Twice a day (morning and evening), zebrafish were fed with normal tetrabit flakes [Tetrabit Gmbh D49304, Melle, Germany; containing crude protein (47.5%), crude fat (6.5%), crude fiber (2.0%), crude ash (10.5%), vitamin A (19.77 IU/g), vitamin D (1.86 IU/g), vitamin E (0.2 mg/g), vitamin C (0.137 mg/g)] and allowed to acclimatize for one week prior to switching with the supplementation of different dietary formulations (as mentioned in Section 2.3).

### 4.3. Formulation of the Specialized Diet and Zebrafish Feeding

The normal tetrabit (ND) was mixed with high cholesterol (4% *w/w*, final) and galactose (30% *w/w*, final) and designated as the HC+HG diet. The HC+HG diet was mixed with banaba leaf extract (BLE, 0.1% *w/w*, final) or policosanol (POL, 0.1% *w/w*, final) or a blend of banaba leaf extract (BLE) and policosanol (POL) (each 0.1% *w/w*, final) to form three different dietary formulations, named as HC+HG+BLE/POL/BLE+POL. A detailed composition and the amount of the different dietary components used to prepare the specialized diet are listed in Appendix A.

Zebrafish (n = 28, group I) were exclusively maintained on the ND diet (control) throughout the study period. Meanwhile other zebrafish (n = 112) were fed an HC+HG diet for two weeks to cause metabolic stress. Post two weeks of supplementation, zebrafish (n = 112) were divided into four groups (n = 28/group) and fed with the designated diet for 12 weeks (Figure 11). Zebrafish in group II were maintained essentially on the HC+HG diet, while the zebrafish in groups III, IV, and V were fed with HC+HG+BLE, HC+HG+POL, and HC+HG+BLE+POL, respectively. Notably, for each group (n = 28), zebrafish were disturbed into four distinct tanks (n = 7/tank) and fed with the specialized diet (70 mg twice) in the morning (~9 a.m.) and evening (~6 p.m.), i.e., a total of 140 mg/tank (~20 mg/zebrafish). During the 12 weeks of the study period, the survivability of zebrafish in each group was recorded every day (week 0–week 12), while the body weight was measured on the starting day (week 0), week 8, and week 12. The time point at week 0 was selected to obtain the baseline body weight, while week 8 served as an intermediate time point, and week 12 was selected to assess the net body weight gain on the final day following exposure to the different dietary formulations. Food consumption efficiency was evaluated in all experimental groups at weeks 0, 8, and 12 to determine the impact of respective diets on zebrafish preference, dislike, and appetite. After 30 min of dietary exposure, consumption was calculated using the formula: [(Total amount of the given food − residual amount of food)/total amount of given food] × 100.

Post-12 weeks of feeding, the average body length (BL, mm) and body depth (BD, mm) of zebrafish in each experimental group were analyzed using ImageJ software (https://imagej.net/ij, version 1.53, assessed on 6 June 2023) to determine the morphometric changes. The morphometric changes are expressed as the ratio of BL/BD.

### 4.4. Zebrafish Euthanasia and Collection of Blood and Organs

Following 12-week supplementation, zebrafish across the groups were deprived of their respective diets (~14 h) prior to the collection of blood. For the collection of blood, zebrafish of a particular group (n = 7/tanks) were euthanized using hypothermic shock [70], and ~2–5 μL of blood/zebrafish was collected and pooled in a single tube and mixed with phosphate-buffered saline (PBS)–ethylenediaminetetraacetic acid (EDTA, 1 mM) at 2:3 (*v/v*) and processed for centrifugation to separate the blood cells and attain the plasma. Different organs (liver, kidney, brain, testes, and ovaries) were surgically removed and preserved in formalin solution (10%).

### 4.5. Quantification of Blood Lipoprotein, Hepatic Function Enzymes, and Glucose Levels

The amount of plasma total cholesterol (TC), triglycerides (TGs), high-density lipoprotein cholesterol (HDL-C), and hepatic function enzymatic biomarkers [aspartate aminotransferase (AST) and alanine aminotransferase (ALT)] was determined using the commercial assay kit, following the methodology suggested by the manufacturers. A detailed methodology is provided as Appendix A. An automated blood glucose meter (AccuCheck, Roche, Basel, Switzerland) was used to determine blood glucose levels.

### 4.6. Blood Oxidative Variables, Ferric Ion Reduction (FRA), and Paraoxonase (PON) Activity

The blood malondialdehyde (MDA) and sulfhydryl content was quantified using the previously described method [71]. In brief, a plasma sample (50 μL, equivalent to 1 mg/mL protein) was mixed with trichloroacetic acid (50 μL, 0.2 mg/μL) and thiobarbituric acid (100 μL, 6.7 μg/μL). Following a 10 min incubation at 95 °C, the absorbance at 560 nm was recorded.

For the estimation of sulfhydryl content, plasma (50 μL, equivalent to 1 mg/mL protein) was mixed with an equal volume of 4 μg/μL 5,5-dithio-bis-(-nitrobenzoic acid), and the content was incubated at room temperature (RT) for 2 h; subsequent absorbance was recorded at 412 nm. The results are expressed as mmol sulfhydryl/mg of protein using the molar absorbance coefficient of the formed product [71].

For the ferric ion reduction ability (FRA), plasma (20 μL, equivalent to 1 mg/mL protein) was mixed with FRA reagent (180 μL) [71], and the content was incubated at RT for 60 min followed by absorbance evaluation at 593 nm.

For the paraoxonase (PON) activity, plasma (40 μL, equivalent to 1 mg/mL protein) was blended with 0.15 mg/μL paraoxon ethyl (160 μL). The mixture was incubated for 2 h at RT, and subsequently, absorbance was detected at 415 nm. The PON activity was expressed as μU/L/min using the molar absorbance coefficient of the formed product [71]. For all the biochemical analyses, four replicates (n = 4) per group were performed, and the results are presented as the mean ± SEM.

### 4.7. Historical Analysis and Immunohistochemistry (IHC)

For the histological analysis, tissue sections (7 μm thick) of different organs were obtained using the cryo-microtome (Leica Biosystem, Nussloch, Germany). Individual sections of liver, brain, kidney, testes, and ovaries were processed for hematoxylin and eosin staining (H&E) [72] to determine the pathological changes in the respective organs.

The accumulation of lipid droplets in liver and testes was examined by oil red O (ORO) staining [70]. In brief, the respective tissue sections (7 μm thick) were covered with an ORO stain solution and incubated at 60 °C for 5 min, followed by washing with 60% isopropanol. After drying, the stained section was examined under a microscope.

The IHC was performed to determine the interleukin (IL)-6 [73] in the liver section and 4-hydroxynonenal (4-HNE) accumulation in the brain. For the IL-6 detection, the 7 μm thick liver section was covered with 200× diluted IL-6-specific antibody (Abcam ab9324, Cambridge, UK). After 16 h of incubation in the humid and cool environment (4 °C), the section was developed using the EnVision+ system HRP polymer kit (Dako, Glostrup, Denmark). For the 4-HNE accumulation, the brain section (7 μm thick) was covered with 200× diluted anti-4-HNE antibody (Abcam ab48506, Cambridge, UK), followed by 16 h of incubation in humid and cool conditions. After incubation, the bound primary 4-HNE antibody was detected using secondary immunoglobulin Alexa Fluor^TM^ 594 (Invitrogen, Carlsbad, CA, USA, AB_2534073). The IHC section was detected under fluorescence microscopy at an excitation wavelength of 590 nm and an emission wavelength of 618 nm.

### 4.8. Dihydroethidium (DHE), Acridine Orange (AO), and Senescent Staining

For dihydroethidium (DHE) and acridine orange (AO) fluorescence staining, the tissue section was covered with 30 μM DHE and 30 μg/mL AO solutions [70]. The section was incubated in the dark for 5 min at RT and visualized under a fluorescence microscope. For the detection of DHE fluorescence intensity, the excitation/emission wavelength of 585/615 nm was used, while for the detection of AO fluorescence intensity, a 505/535 nm excitation/emission wavelength was utilized.

To detect cellular senescence, the 7 μm thick tissue section was entirely submerged with 0.1% of 5-bromo-4-choloro-3-indolyl-β-D-galactopyranoside solution for 16 h [74]. After incubation, the section was washed three times with phosphate-buffered saline (PBS) and visualized under a microscope to detect senescent-positive cells (blue-stained).

### 4.9. Statistical Analysis

For the multiple comparison among the groups, the one-way analysis of variance (ANOVA) was performed, followed by Tukey’s post hoc analysis using SPSS software (version 29, Chicago, IL, USA). The repeated measure ANOVA test was performed after detecting the normal distribution using the Kolmogorov–Smirnov test on the data under the influence of two factors.

## 5. Conclusions

A 12-week dietary intervention of BLE in combination with POL synergistically improved TC, TG, and blood glucose levels disrupted by HC+HG. Also, BLE+POL consumption improved HDL-C levels, reduced oxidative stress markers, and boosted antioxidant parameters. Compared with individual supplementation with BLE or POL, a combined intake of BLE+POL effectively inhibited ROS generation and prevented premature aging of various organs. Specifically, BLE+POL synergistically reduced the buildup of lipid peroxidation species 4-HNE in the brain, protecting against HC+HG-induced apoptosis and cellular senescence. In summary, the study outcome proposes BLE+POL as an effective nutraceutical blend for alleviating metabolic stress, boosting antioxidant capacity and preserving organ health in response to HC+HG-induced events; nevertheless, additional clinical trials are required to confirm the efficacy of BLE+POL in humans.

## Figures and Tables

**Figure 1 ijms-26-07669-f001:**
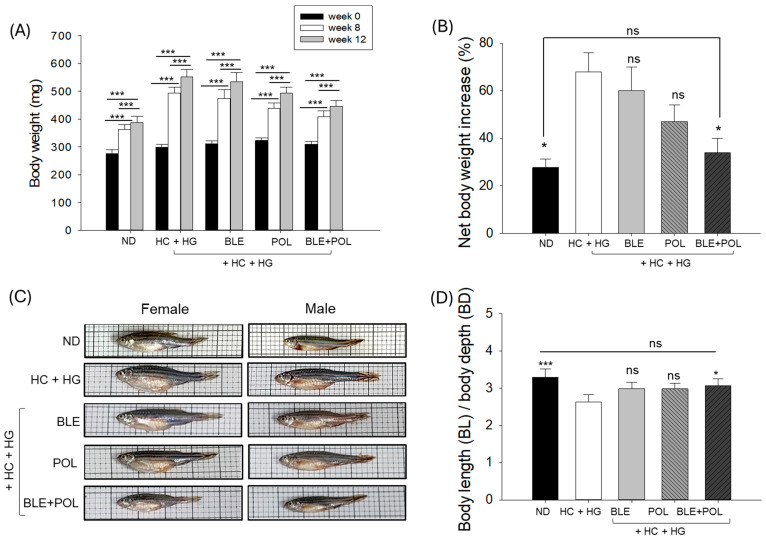
Body weight analysis of zebrafish (**A**) at 0-, 8-, and 12-weeks following intervention of banaba leaf extract (BLE), policosanol (POL), and banaba+policosanol (BLE+POL) infused with a high-cholesterol (HC) and high-galactose (HG) diet. (**B**) Net body weight increases at 12 weeks post-supplementation compared with the baseline value (week 0 body weight). (**C**) Representative images depicting the morphology of female and male zebrafish across the groups post-12 weeks of supplementation of the different dietary formulations. (**D**) Ratio of zebrafish body length (BL) and body depth (BL). ND indicates normal tetrabit diet; HC+HG indicates high-cholesterol (4% *w/w*, final)+high-galactose (30% *w/w*, final) diet. BLE, POL, and BLE+POL indicate banaba leaf extract (0.1% *w/w*, final), policosanol (0.1% *w/w*, final), and banaba leaf extract+policosanol mixed (0.1% *w/w* each) preparation amalgamated with HC+HG. The * (*p* < 0.05), *** (*p* < 0.001) highlights the statistical difference between the marked groups. The “ns” underscores the non-significant difference between the groups.

**Figure 2 ijms-26-07669-f002:**
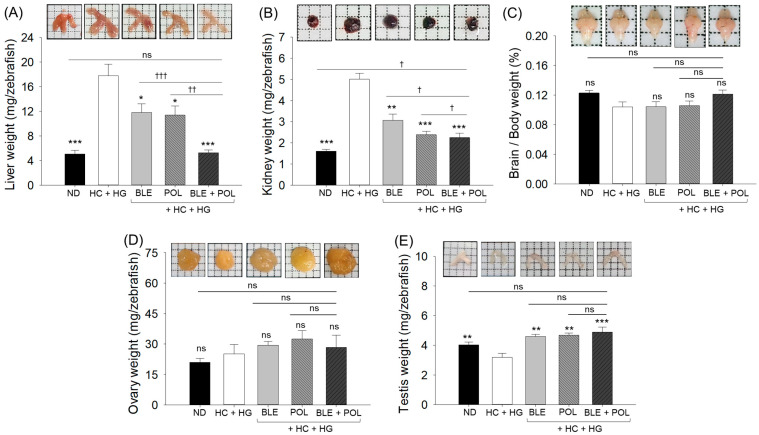
Evaluation of zebrafish organ morphology obtained from the 12-week dietary intervention of banaba leaf extract (BLE), policosanol (POL), and BLE+POL infused with a high-cholesterol (HC) and high-galactose (HG) diet. Organ weight of (**A**) liver, (**B**) kidney, (**C**) brain, (**D**) ovary, and (**E**) testes. ND indicates normal tetrabit diet; HC+HG indicates high-cholesterol (4% *w/w*, final)+high-galactose (30% *w/w*, final) diet. BLE, POL, and BLE+POL indicate banaba leaf extract (0.1% *w/w*, final), policosanol (0.1% *w/w*, final), and banaba leaf extract+policosanol mixed preparation (0.1% *w/w*, each) amalgamated with HC+HG. Statistical differences at *p* < 0.05 (*), *p* < 0.01 (**), and *p* < 0.001 (***) are compared with the HC+HG group, while *p* < 0.05 (^†^), *p* < 0.01 (^††^), and *p* < 0.001 (^†††^) are compared with the BLE+POL group. The “ns” represents the non-significant difference.

**Figure 3 ijms-26-07669-f003:**
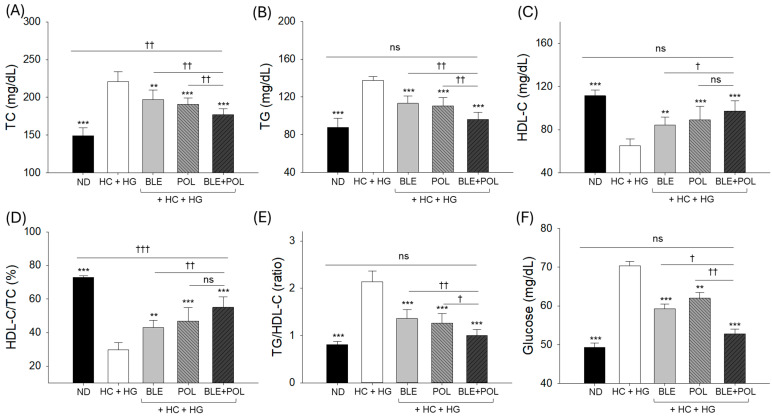
Quantification of (**A**) total cholesterol (TC), (**B**) triglycerides (TGs), (**C**) high-density lipoprotein cholesterol (HDL-C), (**D**) HDL-C/TC (%), (**E**) TG/HDL-C (ratio), and (**F**) glucose levels in the zebrafish blood post-12 weeks of supplementation of banaba leaf extract (BLE), policosanol (POL), and BLE+POL infused with a high-cholesterol (HC) and high-galactose (HG) diet. ND indicates normal tetrabit diet; HC+HG indicates the high-cholesterol (4% *w/w*, final)+high-galactose (30% *w/w*, final) diet. BLE, POL, and BLE+POL indicate banaba leaf extract (0.1% *w/w*, final), policosanol (0.1% *w/w*, final), and banaba leaf extract+policosanol mixed (0.1% *w/w* each) preparation amalgamated with HC+HG. Statistical differences at *p* < 0.01 (**) and *p* < 0.001 (***) are compared with the HC+HG group, while *p* < 0.05 (^†^), *p* < 0.01 (^††^), and *p* < 0.001 (^†††^) are compared with the BLE+POL group. The “ns” represents the non-significant difference.

**Figure 4 ijms-26-07669-f004:**
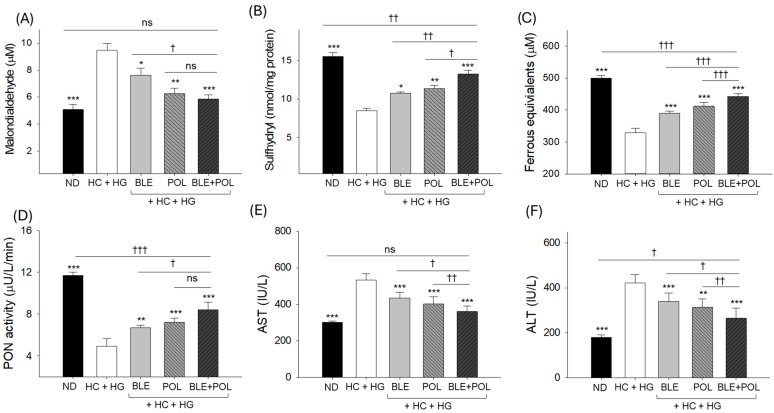
Quantification of (**A**) malondialdehyde level (MDA), (**B**) sulfhydryl content, (**C**) ferric ion reduction (FRA) ability, (**D**) paraoxonase (PON) activity, (**E**) aspartate aminotransferase (AST), and (**F**) alanine aminotransferase (ALT) levels in the zebrafish blood from the 12-week supplementation of banaba leaf extract (BLE), policosanol (POL), and BLE+POL infused with a high-cholesterol (HC) and high-galactose (HG) diet. ND indicates normal tetrabit diet; HC+HG indicates the high-cholesterol (4% *w/w*, final)+high-galactose (30% *w/w*, final) diet. BLE, POL, and BLE+POL indicate banaba leaf extract (0.1% *w/w*, final), policosanol (0.1% *w/w*, final), and banaba leaf extract+policosanol mixed (0.1% *w/w* each) preparation amalgamated with HC+HG. Statistical differences at *p* < 0.05 (*), *p* < 0.01 (**), and *p* < 0.001 (***) are compared with the HC+HG group, while *p* < 0.05 (^†^), *p* < 0.01 (^††^), and *p* < 0.001 (^†††^) are compared with the BLE+POL group. The “ns” represents the non-significant difference.

**Figure 5 ijms-26-07669-f005:**
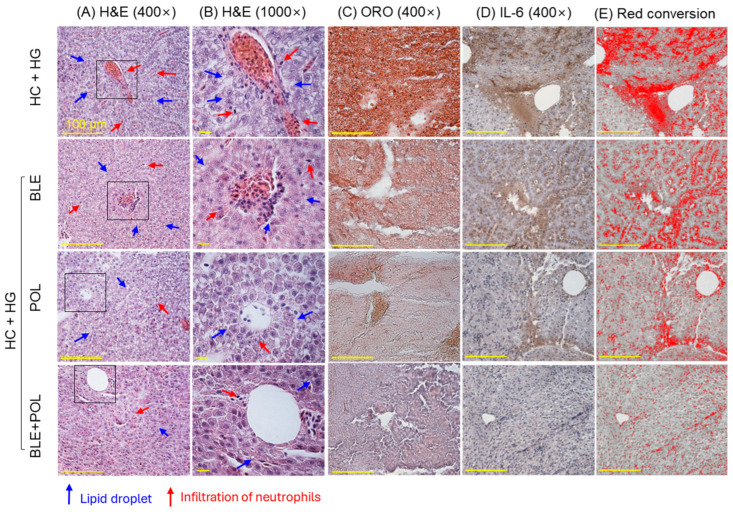
Histological analysis of liver (**A**) and (**B**) hematoxylin and eosin (H&E) staining at 400× and 1000× magnification, (**C**) oil red O (ORO) staining, (**D**) immunohistochemical (IHC) images for interleukin (IL)-6, (**E**) red conversion of the brown colored IL-6-stained area at the brown color threshold value (20–120) using Image J software (version 1.53, https://imagej.net/ij, assessed on 6 June 2023). The red conversion was performed to enhance the visibility. [Scale bar = 100 μm]. (**F**–**H**): Quantification of neutrophil counts, ORO, and IL-6-stained area, respectively. All the images were taken from the liver excised from the 12-week dietary intervention of banaba leaf extract (BLE), policosanol (POL), and BLE+POL infused with a high-cholesterol (HC) and high-galactose (HG) diet. HC+HG indicates the high-cholesterol (4% *w/w*, final)+high-galactose (30% *w/w*, final) diet. BLE, POL, and BLE+POL indicate banaba leaf extract (0.1% *w/w*, final), policosanol (0.1% *w/w*, final), and banaba leaf extract+policosanol mixed (0.1% *w/w* each) preparation amalgamated with HC+HG. Statistical differences at *p* < 0.05 (*), *p* < 0.01 (**), and *p* < 0.001 (***) are compared with the HC+HG group, while *p* < 0.05 (^†^), and *p* < 0.01 (^††^) are compared with the BLE+POL group. The “ns” represents the non-significant difference.

**Figure 6 ijms-26-07669-f006:**
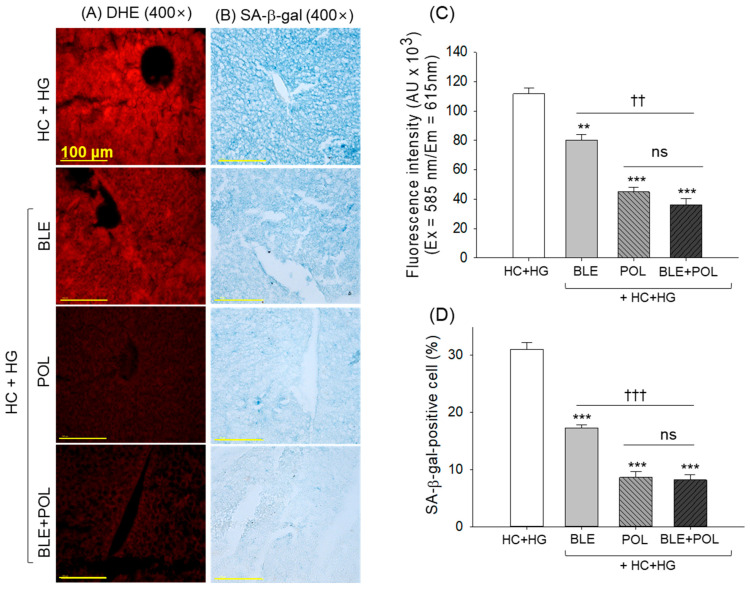
Liver (**A**) dihydroethidium (DHE) and (**B**) senescent-associated β-galactosidase (SA-β-gal) staining. The images were taken from the liver excised from the 12-week dietary intervention of banaba leaf extract (BLE), policosanol (POL), and BLE+POL infused with a high-cholesterol (HC) and high-galactose (HG) diet. [Scale bar = 100 μm]. (**C**,**D**): Quantification of DHE and SA-β-gal-stained area, respectively. HC+HG indicates the high-cholesterol (4% *w/w*, final)+high-galactose (30% *w/w*, final) diet. BLE, POL, and BLE+POL indicate banaba leaf extract (0.1% *w/w*, final), policosanol (0.1% *w/w*, final), and banaba leaf extract+policosanol mixed (0.1% *w/w* each) preparation amalgamated with HC+HG. Statistical differences at *p* < 0.01 (**) and *p* < 0.001 (***) are compared with the HC+HG group, while *p* < 0.01 (^††^) and *p* < 0.001 (^†††^) are compared with the BLE+POL group. The “ns” represents the non-significant difference.

**Figure 7 ijms-26-07669-f007:**
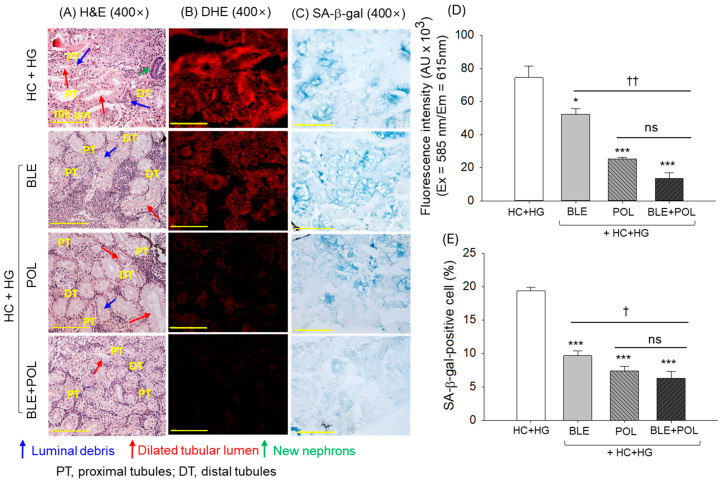
Histological analysis of zebrafish kidney (**A**) hematoxylin and eosin (H&E) staining, (**B**) dihydroethidium (DHE), and (**C**) senescent-associated-β-galactosidase (SA-β-gal) staining. Qualification of (**D**) DHE fluorescence intensity and (**E**) SA-β-gal-positive cells. [Scale bar = 100 μm]. The images were taken from the kidney excised from the 12-week dietary intervention of banaba leaf extract (BLE), policosanol (POL), and BLE+POL infused with a high-cholesterol (HC) and high-galactose (HG) diet. HC+HG indicates the high-cholesterol (4% *w/w*, final)+high-galactose (30% *w/w*, final) diet. BLE, POL, and BLE+POL indicate banaba leaf extract (0.1% *w/w*, final), policosanol (0.1% *w/w*, final), and banaba leaf extract+policosanol mixed (0.1% *w/w* each) preparation amalgamated with HC+HG. Statistical differences at *p* < 0.05 (*) and *p* < 0.001 (***) are compared with the HC+HG group, while *p* < 0.05 (^†^) and *p* < 0.01 (^††^) are compared with the BLE+POL group. The “ns” represents the non-significant difference.

**Figure 8 ijms-26-07669-f008:**
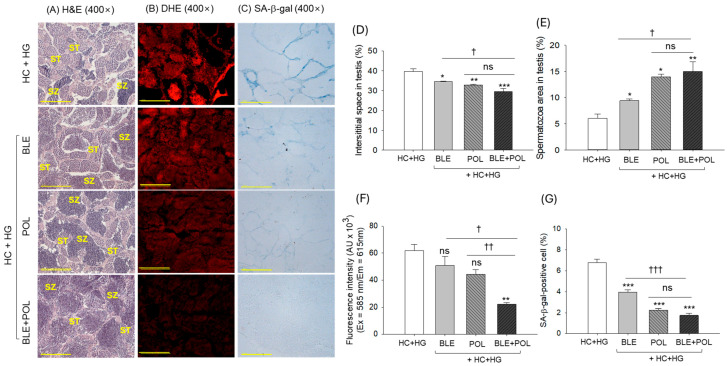
Histological analysis of zebrafish testes (**A**) hematoxylin and eosin (H&E) staining, (**B**) dihydroethidium (DHE), and (**C**) senescent-associated-β-galactosidase (SA-β-gal) staining. Qualification of (**D**) interstitial space between seminiferous tubules, (**E**) spermatozoa area, (**F**) DHE fluorescence intensity, and (**G**) SA-β-gal-stained area. ST and SZ are abbreviated for spermatocytes and spermatozoa, respectively. [Scale bar = 100 μm]. The images were taken from the testes excised from the 12-week dietary intervention of banaba leaf extract (BLE), policosanol (POL), and BLE+POL infused with a high-cholesterol (HC) and high-galactose (HG) diet. HC+HG indicates the high-cholesterol (4% *w/w*, final)+high-galactose (30% *w/w*, final) diet. BLE, POL, and BLE+POL indicate banaba leaf extract (0.1% *w/w*, final), policosanol (0.1% *w/w*, final), and banaba leaf extract+policosanol mixed (0.1% *w/w* each) preparation amalgamated with HC+HG. Statistical differences at *p* < 0.05 (*), *p* < 0.01 (**), and *p* < 0.001 (***) are compared with the HC+HG group, while *p* < 0.05 (^†^), *p* < 0.01 (^††^), and *p* < 0.001 (^†††^) are compared with the BLE+POL group. The “ns” represents the non-significant difference.

**Figure 9 ijms-26-07669-f009:**
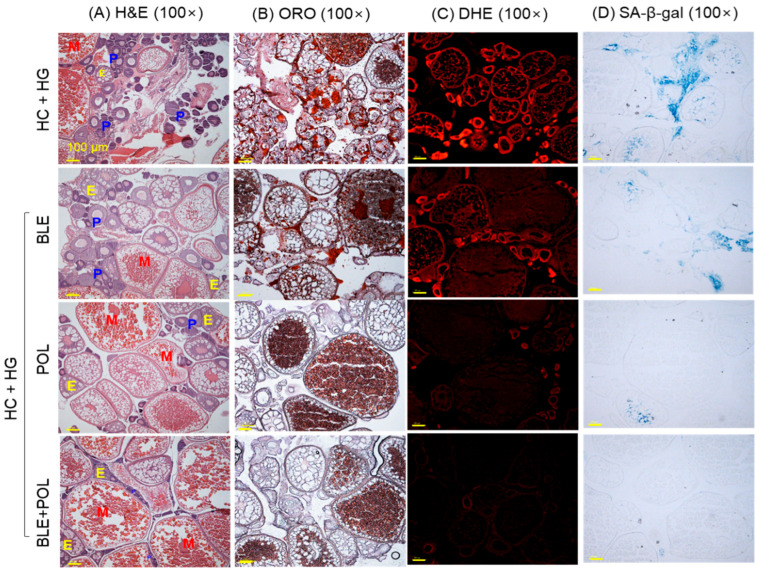
Histological analysis of zebrafish ovary (**A**) hematoxylin and eosin (H&E) staining, (**B**) oil red O (ORO), (**C**) dihydroethidium (DHE), and (**D**) senescent-associated-β-galactosidase (SA-β-gal) staining. Qualification of (**E**) pre-vitellogenic, (**F**) early-vitellogenic, and (**G**) mature-vitellogenic oocytes. M, P, and E are abbreviated for mature, pre-, and early vitellogenic oocytes. [Scale bar = 100 μm]. (**H**–**J**): Quantification of ORO-stained area, DHE fluorescence intensity, and SA-β-gal-stained area, respectively. The images were taken from the ovaries excised from the 12-week dietary intervention of banaba leaf extract (BLE), policosanol (POL), and BLE+POL infused with a high-cholesterol (HC) and high-galactose (HG) diet. HC+HG indicates the high-cholesterol (4% *w/w*, final)+high-galactose (30% *w/w*, final) diet. BLE, POL, and BLE+POL indicating banaba leaf extract (0.1% *w/w*, final), policosanol (0.1% *w/w*, final), and banaba leaf extract+policosanol mixed (0.1% *w/w* each) preparation amalgamated with HC+HG. Statistical differences at *p* < 0.01 (**) and *p* < 0.001 (***) are compared with the HC+HG group, while *p* < 0.05 (^†^) and *p* < 0.01 (^††^) are compared with the BLE+POL group. The “ns” represents the non-significant difference.

**Figure 10 ijms-26-07669-f010:**
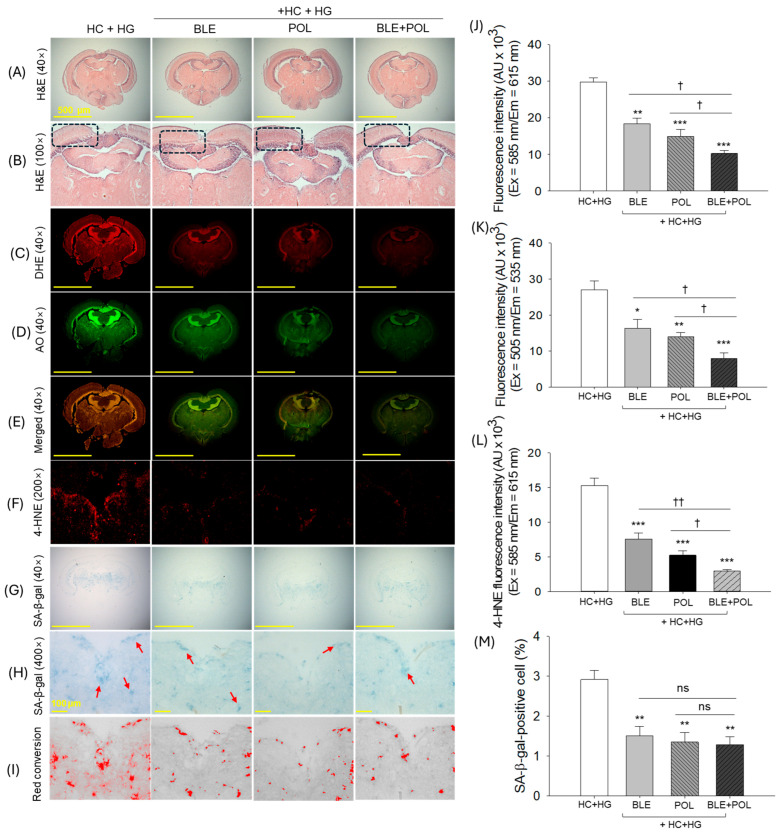
Histological analysis of zebrafish brain (**A**,**B**) hematoxylin and eosin (H&E) staining at 40× and 100× magnification. The magnified view of the images inside the dotted box is depicted in Appendix A. (**C**) Dihydroethidium (DHE), (**D**) acridine orange (AO), (**E**) merged image of DHE- and AO-stained area, (**F**) 4-hydroxynoenal (4-HNE). (**G**,**H**): senescent-associated-β-galactosidase (SA-β-gal) staining at 40× and 100× magnification, red arrow highlights the senescent positive cells. (**I**) Red conversion of SA-β-gal stained area using Image J software (at blue color threshold value of 0–120) to improve the visibility. Qualification of (**J**) DHE, (**K**) AO, and (**L**) 4-HNE fluorescence intensity and (**M**) SA-β-gal-stained area. The images were taken from the brain excised from the 12-week dietary intervention of banaba leaf extract (BLE), policosanol (POL), and BLE+POL infused with a high-cholesterol (HC) and high-galactose (HG) diet. HC+HG indicates the high-cholesterol (4% *w/w*, final)+high-galactose (30%, *w/w* final) diet. BLE, POL, and BLE+POL indicating banaba leaf extract (0.1% *w/w*, final), policosanol (0.1% *w/w*, final), and banaba leaf extract+policosanol mixed preparation (0.1% *w/w* each) amalgamated with HC+HG. Statistical differences at *p* < 0.05 (*), *p* < 0.01 (**), and *p* < 0.001 (***) are compared with the HC+HG group, while *p* < 0.05 (^†^) and *p* < 0.01 (^††^) are compared with the BLE+POL group. The “ns” represents the non-significant difference.

**Figure 11 ijms-26-07669-f011:**
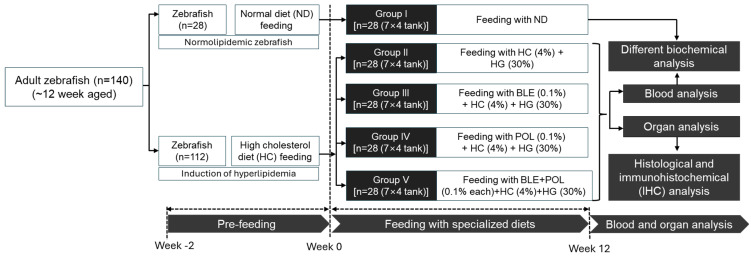
Experimental layout and the study plan. ND indicates the normal tetrabit diet. HC+HG indicates the high-cholesterol (4% *w/w*, final)+high-galactose (30% *w/w*, final) diet. BLE, POL, and BLE+POL indicate banaba leaf extract (0.1% *w/w*, final), policosanol (0.1% *w/w*, final), and banaba leaf extract+policosanol mixed preparation (0.1% *w/w* each) amalgamated with HC+HG.

## Data Availability

The data used to support the findings of this study are available from the corresponding author upon reasonable request.

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
