# Peer review of "Co-Supplementation of Policosanol and Banaba Leaf Extract Exhibited a Cooperative Effect Against Hyperglycemia and Dyslipidemia in Zebrafish: Highlighting Vital Organ Protection Against High-Cholesterol and High-Galactose Diet"

_ijms, 2025, doi:10.3390/ijms26167669_

Round 1

Reviewer 1 Report

Comments and Suggestions for Authors

This is a very sophisticated, advanced and complex study analyzing the effect of policosanol and banaba leaf extract supplementation on hyperglycemia and dyslipidemia in zebrafish. Introduction justifies the study, methods which were applied are appropriate, results are almost clearly presented and well discussed. Conclusions briefly summarize the obtained results and indicate future research needs. Dear Authors, congratulations of your study! I was really impressed by it, as I do several reviews a year of a similar research this one impressed me a lot.
The only remark which I have concerns Figure 2D - something's wrong in y axis description - it's really hard to believe that the percentage of ovary mass related to body weight is so high.

Author Response

Thank you for your insightful comments. Following the reviewer’s suggestion, we made point-to-point response and reflected on revision.

Please find attached doc as our response.

Reviewer 2 Report

Comments and Suggestions for Authors

The motivation for the manuscript is unclear. Lipid metabolism disorders can be caused by 1. genetic factors, 2. lifestyle, and 3. concomitant diseases. It is obvious that new horizons for possible future therapy are suitable for cases associated with lipid metabolism disorders caused by poor nutrition, but will this work with bad habits, such as a sedentary lifestyle, smoking, and alcohol abuse? Can the model developed by the authors help with lipid metabolism disorders caused by type 2 diabetes or hypothyroidism? Fatty liver disease, chronic hepatitis? Nephrotic syndrome? The authors need to clearly formulate the motivation for their research!

My personal experience suggests that the main cause of acquired dyslipidemia is the lack of self-discipline and self-control. In essence, the authors propose to treat not the cause, but the effect. It seems to me that this should also be reflected in the introduction chapter of the manuscript.

I also have serious doubts that zebra fish are an adequate model for induced dyslipidemia. In my opinion, zebrafish feed on small insects, larvae, algae, plankton, and plant seeds in the water in the wild. There are no and have never been any monosaccharides or cholesterol-rich foods in the diet. The authors need to respond to this comment in the introduction section of the manuscript, since this train of thought is formed in the mind of any fish lover... 

All the graphs include a case with a high cholesterol (HC) and galactose (HG) diet. There is a case when banaba leaf extract (BLE) or policosanol (POL) + their combination are used against the background of this diet. Everything is considered in the time interval of 0, 8, 12 weeks. By the way, it would not be superfluous to describe in more detail in the manuscript why these particular times were chosen. Everything would seem to be fine! But I don’t think so, the manuscript lacks control! Namely, a reference, a healthy organism that did not receive huge doses of HC+HG. Control values are needed. These values will show readers and authors how close can get to the physiological norm.

Mixtures of long-chain fatty alcohols were used in the USSR, China and a number of Eastern European countries to correct diabetes. Is this close to the data obtained in this manuscript? Can the authors comment on this somehow?

The photographs of the fish in Figure 1c can provide a lot of information. Can the authors conduct computer morphometry of the fish body parameters? If this is difficult to do, can the authors at least enlarge the photographs so that morphometry can be done independently?

Acquired dyslipidemia is a disorder of the lipid spectrum of the blood, which is manifested by an increase in cholesterol, triglycerides, low and very low density lipoproteins and a decrease in high density lipoproteins, or alpha-lipoproteins. That is, the observed changes are observed in the bloodstream. Why is most of the work presented in the manuscript carried out on internal organs? Why do the authors not study the cellular composition of the blood? The formation of plaques on blood vessels? This is the simplest and most convincing!

Overall, the study is quite well thought out. The manuscript is well illustrated, histology and biochemical markers are compared. The materials will certainly be of interest to readers. I hope my comments will help improve the quality of the manuscript text.

Author Response

(The authors gave the same response as above.)

Reviewer 3 Report

Comments and Suggestions for Authors

The important finding reported in the manuscript is that a combination of banaba leaf extract and policosanol can effectively address issues related to dyslipidemia and elevated blood glucose levels. Some blood parameters, including the level of two aminotransferases, are reduced because of the combined intake of two mentioned products of natural origin. Data on combined formulation of these two products is scarce, so the material of manuscript fills a certain gap.

Policosanol, a mixture of long-chain fatty alcohols obtained from sugarcane wax represents a valuable biologically active product of plant origin.

In the introduction part, which is very informative, authors provide data on the danger of dyslipidemia, prospects of policosanol as medicine as well as huge potential of banaba leaves as an antidiabetic agent.

Experimental issues are presented and discussed logically; a 12-week dietary intervention was performed. Blood parameters and all other issues are measured with modern equipment (cryo-microtome et al), body weigh analysis is perfect, organ morphology of liver is studied carefully, lipoprotein profile and histological analysis of a set of tissues are described in detail, Zebrafish culturing is reported, dietary components  and details of feeding in the process of investigation are described, data proceeding (statistical analysis et al) is reliable.

The results of study are interesting to wide audience.

The quality of the pictures is good; references are sufficient, and the conclusions are convincing.

The manuscript can be published after minor revision:

  • Abbreviation ANOVA is used in lines 93, 94 without explanation, and only in the line 649 a reader can get that it is “the one-way analysis of variance”.
  • Ref 68: not bold the number of ref, but year should be in bold.

Author Response

(The authors gave the same response as above.)

Round 2

Reviewer 2 Report

Comments and Suggestions for Authors

I hope my comments helped to make the manuscript better. I recommend it for publication.